# Estimating District-Level Electricity Consumption Using Remotely Sensed Data in Eastern Economic Corridor, Thailand

**Sirikul Hutasavi \* and Dongmei Chen**

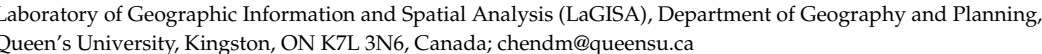

Laboratory of Geographic Information and Spatial Analysis (LaGISA), Department of Geography and Planning, Queen's University, Kingston, ON K7L 3N6, Canada; chendm@queensu.ca
* Correspondence: 16sh48@queensu.ca; Tel.: +1-343-363-6518

**Abstract:** The intensive industrial development in special economic zones, such as Thailand's Eastern Economic Corridor, increases energy consumption, leading to an imbalance of energy supply and a challenge for energy management. Electricity consumption at a local level is crucial for utility planners to manage and invest in the electrical grid. With this study, we propose an electricity consumption estimation model at the district level using machine learning with publicly available statistical data and built-up area (BU), area of lit (AL), and sum of light intensity (SL) data extracted from Landsat 8 and Suomi NPP satellite nighttime light images. The models created from three machine learning algorithms, which included Multiple Linear Regression (MR), Decision Tree (DT), and Support Vector Regression (SVR), were compared. The results show that (1) electricity consumption is highly correlated with SL, AL, and BU; and (2) the DT model demonstrated a better performance in predicting local electricity consumption when compared to MR and SVR with the lowest error rate and highest $R^2$. The local government in developing countries with limited data and financial resources can adopt the proposed approach to benefit from utilizing commonly available remote sensing and statistical data with simple machine learning models such as DT (regression method) for sustainable electricity management.

**Keywords:** electricity consumption; locally prediction model; remote sensing; decision tree; regression method; machine learning; nighttime imagery

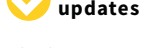



## 1. Introduction

Nowadays, energy has become an essential support for economic growth and is a rapid indicator of human living standards. Electricity is one of the primary energy sources and has a solid relationship with the degree of country development and quality of life in modern society [1]. Energy consumption can be explained by many factors, including economic growth, non-agricultural GDP, population, and urbanization [2]. Consequently, many pieces of previous research have focused on how energy consumption relates to Human Development Index (HDI) at the national level in developed countries [3–5]. These studies aim to sustain the welfare of developed countries through energy demand and its footprint. At the local scale, electricity consumption plays a significant role for utility planners to forecast load and invest in the electrical grid. Therefore, estimating electricity consumption enhances the understanding of human activities in the region and helps authorities determine the appropriate electricity supply [6].

The electricity consumption can be estimated by combining socioeconomic indicators as well as population density, human mobility, and electronic devices per household to predict electricity consumption [6–8]. However, the in-depth electricity load curve data are often unavailable in developing countries, such as Thailand. In such a situation, other proxies such as nighttime light (NTL) data from satellite images and geo-informatic and socioeconomic data are widely applied to estimate electricity load and consumption at the local level. For example, NTL has been applied as a proxy for an electricity-related

indicator such as electricity demand [9], household electrification [10], and monitoring of electricity consumption at different spatial and temporal scales in developing (low-middle income) countries, such as China, India, and others [11]. NTL was founded to be an essential indicator of human activities and settlements (urbanization) in spatial dimensions [12]. In addition, spatial information, as well as land-use types, were also utilized to classify the consumer types for forecasting energy consumption when a history of consumption was not available [13]. Mei et al. [6] proposed to use socio-demographic information (demographic, economic, and sociological statistics at a local scale) as an essential supplement for spatially estimating electrical consumption from source zones to target zones with no or little historical consumption data.

Various prediction techniques have been used to estimate electricity consumption with the aforementioned proxies such as Linear Regression, Spatial Regression, and Machine Learning Algorithms (MLAs). MLA-based prediction models can identify patterns and react to changes in time series data which are essential for decision makers to understand trends. The models are trained and re-trained to generate a final model that builds upon optimizing the desired objective to predict its output. Moreover, MLAs are suited for complex problems because they emphasize model testing and adjusting [14]. Tso and Yau [15] compared Regression Analysis with two MLAs, including Decision Tree (DT) and Neural Network (NN), for forecasting the energy consumption of data centers based on weather conditions. They concluded that DT and NN models performed slightly better than traditional regression models and more complex predictors. In addition, Least Squares Support Vector Machines (LS-SVM) were suggested for accurate and quick prediction of electricity consumption in Turkey [16]. In a study conducted in France, the Spatial–Temporal Model provided excellent performance when estimating electricity consumption at a small temporal scale with socio-demographic and client information [6].

Additionally, Xiao et al. [17] developed a Spatial–Temporal Geographically Weighted Regression Model to simulate electricity consumption in China at the provincial level and achieved goodness of fit of 99% and relative error of less than 5%. The MLAs with a linear regression approach showed satisfactory accuracy in energy consumption forecasting at data centers based on weather information from remotely sensed data [18]. Furthermore, the new hybrid Convolutional Neural Network (CNN) model illustrated excellent performance in forecasting residential and commercial buildings' electricity consumption estimation [19]. Yue et al. [20] integrated the nighttime light data with the spatial econometric model to quantify energy consumption in China at a regional level. Their result illustrated that the Spatial Durbin Model (SDM) provided high accuracy of electricity consumption at provincial and prefectural scales. However, these previous studies focused on estimating the electricity consumption either at a very coarse country level or a very detailed property scale [21]. The countrywide value is impractical for local management, while the fine-scale value requires intensive inputs that are not readily available in many developing nations. Therefore, research on estimating the electricity consumption at a medium scale (district level) remains limited but should be pursued to support the local electricity supply management and quantify the socio-economic impact of intensive industrial development policy (i.e., human well-being).

Thailand's Eastern Economic Corridor (EEC) includes Rayong, Chonburi, and Chachoengsao provinces (Figure 1) and is a particular example representing how the developing country use policy strategy to drive regional development. The EEC development policy aims to create opportunities for investors to participate in developing utilities, infrastructure, and public transportation. Thailand's government expected that EEC development would raise the Gross Provincial Product (GPP) and the country's economy and improve the quality of life of people who lives in the vicinity by connecting the emerging global economy. With the country's rapid development and intensive promotion of the industrial economy, the demand for electricity consumption has increased. The total electricity consumption in the domestic sector rose from 25,525 GWh in 2012 to 29,074 GWh in 2018 [22]. Concurrently, Thailand must diversify the domestic energy source, improve

energy efficiency, and promote community participation in energy management, following Sustainable Development Goal (SDG) 7 to ensure access to affordable, reliable, sustainable, and modern energy for all. Moreover, the local governments of the key district (Rayong, Chonburi, and Chachoengsao provinces) play an essential role in managing the limited resources of electricity to ensure the EEC's sustainable growth.

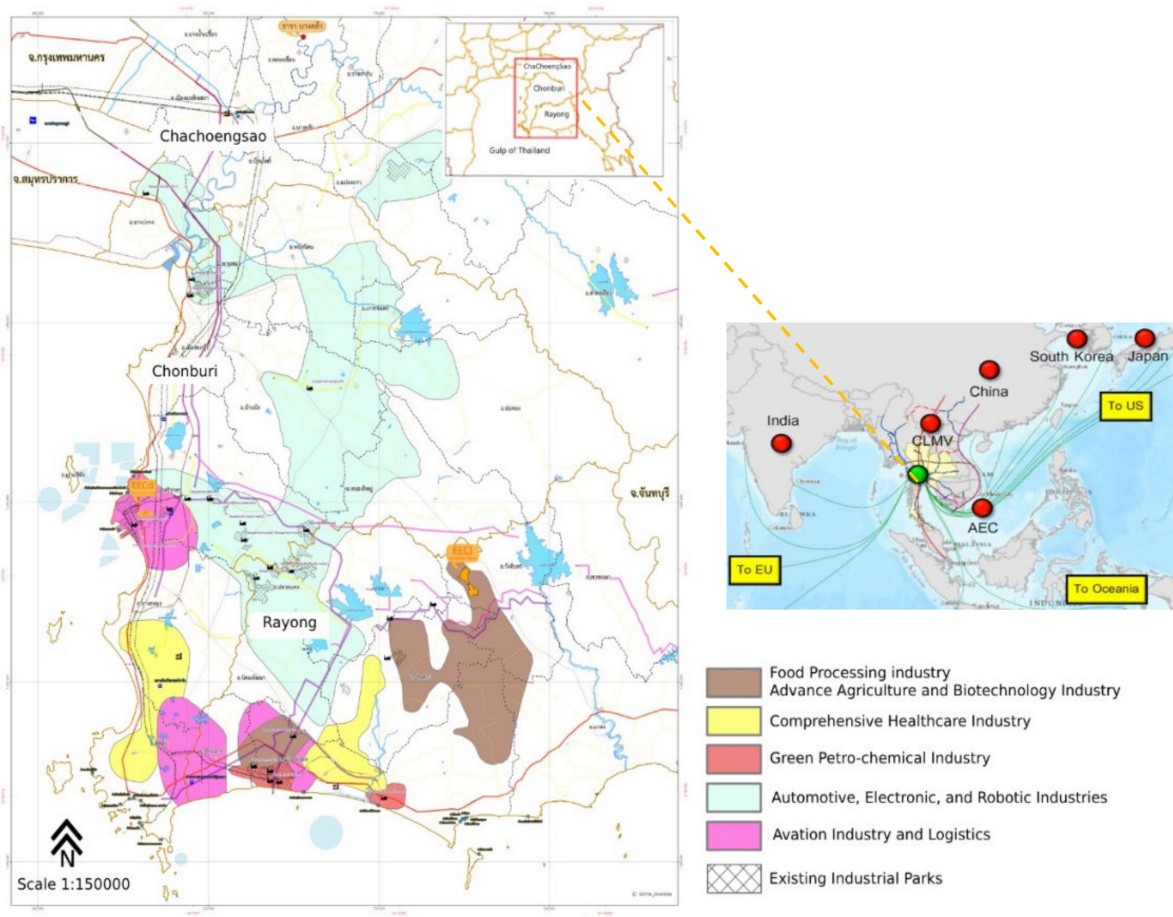

**Figure 1.** The study site (Eastern Economic Corridor, Thailand).

The EEC's electricity consumption is heterogeneous due to the differences in geographic and socioeconomic factors. Rayong and Chonburi are the top GPP generators in Thailand because of their early industrial and tourism development from the Eastern Seaboard policy in 1990. Conversely, Chachoengsao has different economic activities, mainly focusing on agricultural and cultural tourism with medium GPP. Therefore, the electricity consumption estimation at a medium scale (district level) is required to support local government decision making.

In 2015, Phamornchantaramast S. et al. forecasted electricity demand in Chonburi province based on classified land-use types from Landsat 5 and 7. They used the built-up area to estimate the electricity demand per square kilometer. Their result showed that a medium correlation of 0.47 existed between predicted electricity demand and actual electricity consumption [23]. The electricity consumption in the districts close to industrial and tourism development areas was underestimated. Using Thailand's EEC as a case study, this paper aimed to build a suitable estimation model for electricity consumption at the district level by employing and comparing three MLAs (MR, DT, and SVM). Due to the limitation of the electricity load and electricity consumption database within the study area, we extracted nighttime light and built-up area from the remotely sensed data. We used them along with general socioeconomic data, such as population, household, and

population density, as predictors of electricity consumption for 30 districts in the chosen region, based on the number of available data points. The detailed methodology, including dataset, pre-processing, prediction models, and validation approaches, was introduced in the next section. The results were then presented, and the findings were discussed. In the end, a conclusion was provided.

## 2. Materials and Methods

### 2.1. Study Flowchart

The flowchart in Figure 2 presents the processing steps used in this study. First, the remote sensing data were organized through Google Earth Engine (GEE) platform and processed to extract the built-up area, annual average light intensity, and area of lit from Landsat 8 Operational Land Imager (OLI), Suomi NPP Visible Infrared Imaging Radiometer Suite (VIIRS) Day and Night Band (DNB), respectively. Second, the sum of light intensity (SL) using zonal statistic tools was calculated for each district through ArcGIS online. All datasets were then prepared for model building after cleansing and normalizing. The electricity consumption estimation models included MR, DT, and SVR were conducted in RapidMiner® software. The feature selection and the optimization of model parameters were also controlled.

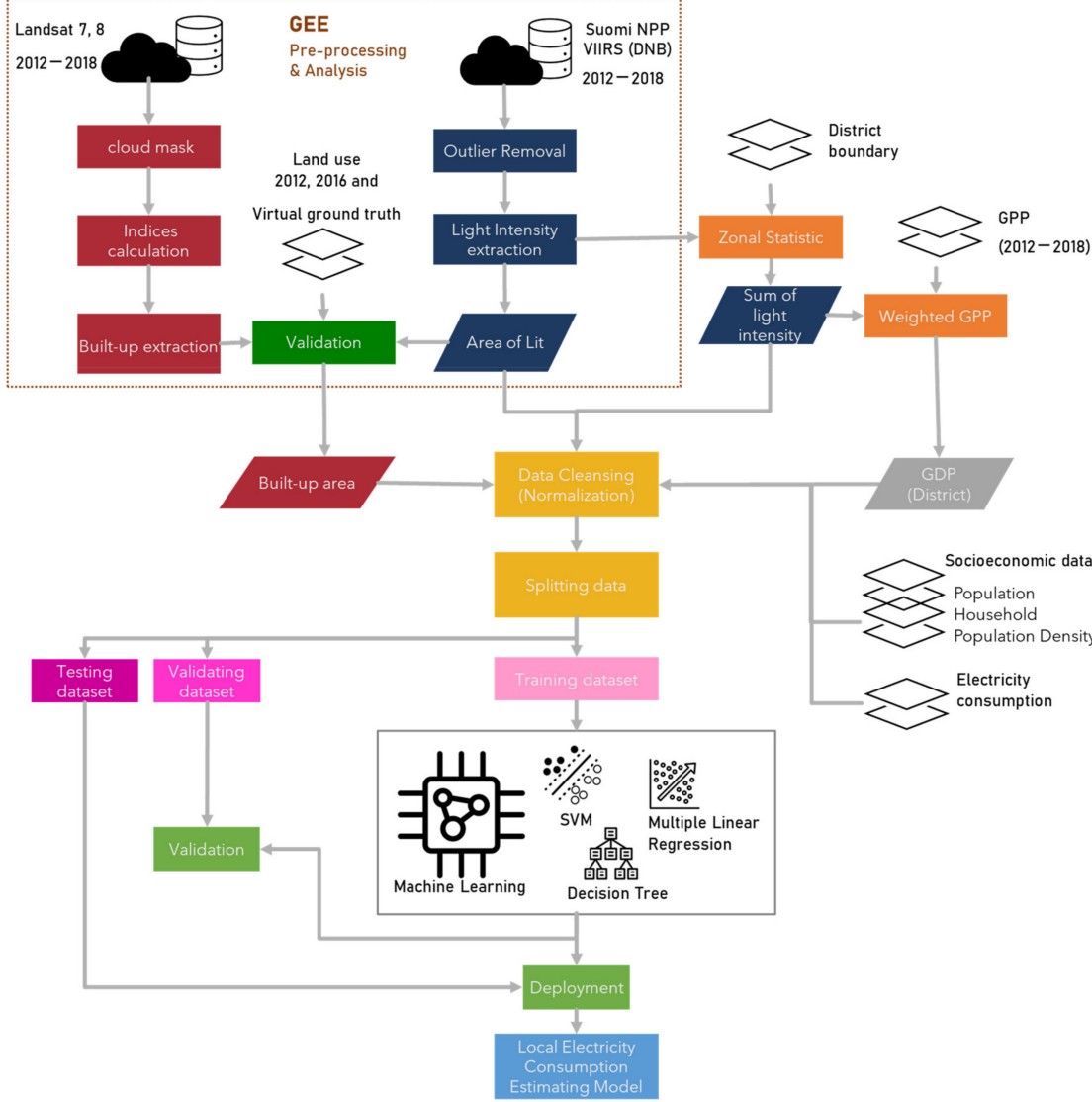

**Figure 2.** Flowchart of the study.

*2.2. Data*

2.2.1. Socioeconomic Data

The National Statistic Organization (NSO) of Thailand provides the general statistic from country to subdistrict level. The population and household data at the district level within EEC regions were downloaded from their website. Pre-processing was then performed to calculate and round up the population density for each district. The number of households showed a gradual increase over time. The population growth is radically fluctuating because of the intensive mobility of both industry and tourism laborers in the EEC's industrial regions. The commuter population increased from 800,000 people in 2010 to 1,000,000 people in 2019, influencing urbanization and the energy consumption rate [24]. The population and households dramatically increased from 2017 to 2018 after implementing a special economic development policy in 2016 [25]. GPP has been regarded as a strong indicator of electricity consumption [21]. We then gathered the 2012 to 2018 Gross Provincial Product per Capita from the website of the Office of the National Economic and Social Development Council (NESDB), Thailand.

2.2.2. Remotely Sensed Data

This study used NTL data extracted from Visible Infrared Imaging Radiometer Suite Day-Night Band (VIIRS-DNB) products and Landsat 7 and 8 ETM images from 2012 to 2018. The time series data were processed on Google Earth Engine (GEE, https://earthengine.google.com accessed on 24 June 2021), a cloud-based computing platform that allows users to conduct geospatial data analysis by providing various computation, analyzation, and visualization. GEE becomes an effective and constructive tool with various satellite images and geospatial datasets [26,27]. GEE demonstrated high potential in the large-scale temporal urbanization analysis process. Pu et al. [27] utilized Landsat time series data and computation algorithms in GEE to map urban areas and achieved an accuracy of 0.8 to 0.9.

In this study, we adopted a method from previous research with processes based on the GEE platform. These include (1) extracting light intensity and area of lit from the annual mean composite of monthly VIIRS-DNB, (2) computing built-up indices from the annual median composition of cloud-masked Landsat 7 and 8 Surface Reflectance (SR) datasets, and (3) classifying the built-up areas from the derived indices using adaptive thresholding method (Otsu's threshold) [28]. The built-up area extraction accuracy of 2012 and 2016 was assessed using 500 random sampling points from the national land-use dataset and virtual ground truth from Landsat 7 and 8 images each year. The overall accuracy of the build-up areas for 2012 and 2016 were 78% and 85% with kappa index values at 74% and 82%, respectively.

The Landsat 7 (2012) and Landsat 8 (2013–2018) dataset was requested from the GEE platform, filtering by year, processing cloud mask, and selecting band 1 to 7. Each year of Landsat data has been reduced by median (derived from a histogram) for pre-compute the Landsat indices. We then adopted the built-up area extraction method from the previous paper, using the Modified Built-up Index (MBUI), which employs Otsu's thresholding technique, and integrates with the nighttime light data [29]. The MBUI is calculated by Equation (1).

$$MBUI = BUI - MNDWI, \tag{1}$$

where:

$BUI = NDBI - NDVI,$
$NDBI = \frac{(SWIR-NIR)}{(SWIR+NIR)},$
$NDVI = \frac{(NIR-Red)}{(NIR+Red)},$ and
$MNDWI = \frac{(Green-SWIR)}{(Green+SWIR)}.$

Considering the histogram of MBUI that represents some bimodal pattern, we choose Otsu's method to classify the built-up area. Otsu's thresholding is a fast and straightforward method to determine an optimal threshold value from an image histogram. The optimal threshold is given when the between-class variance is maximized [30]. Otsu's thresholding

has been used in several pieces of research [31–33]. The sample of the result is presented in Figure 3c.

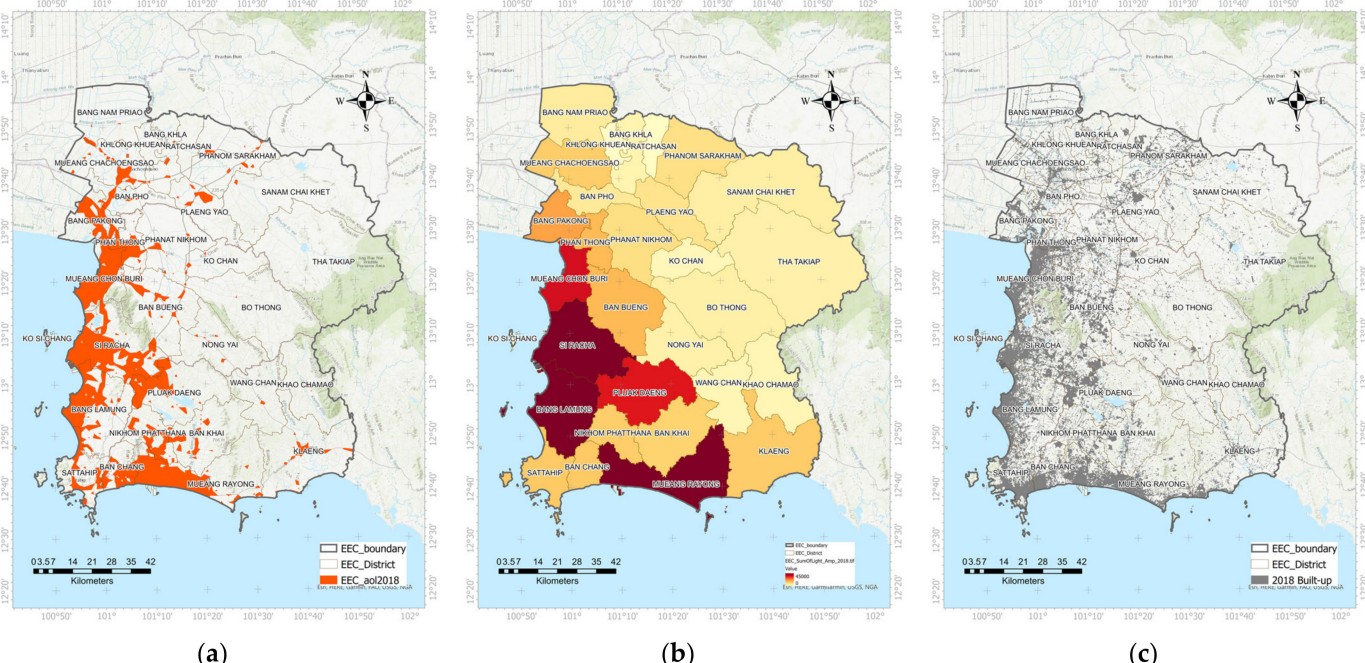

**Figure 3.** The examples of remotely sensed data extraction (Landsat 8 and NPP VIIRS DNB) of 2018 include: (**a**) area of lit, (**b**) sum of light intensity (nW cm$^{-1}$sr$^{-1}$), and (**c**) built-up area.

We employed GEE algorithms to request and export the monthly VIIRS-DNB dataset (2012–2018), which removes cloud cover using the VIIRS cloud mask product (VCM) and correct for stray light [34]. We calculated the annual average light intensity for each year, except the year 2012, which we calculated using the average light intensity from six-monthly datasets (June to December 2012) based on data availability. Then, we adopted the zonal statistic tool in ArcGIS Pro to identify the sum of light intensity in each district polygon from light intensity images. Zonal Statistics is a tool to calculate each zone defined by a zone dataset, based on values from another value raster dataset. A single output value (the sum of light intensity) was computed for every zone (district).

Furthermore, the lit area was extracted using the thresholding method to differentiate each district's dark and bright areas. The threshold value was calculated for each year from 2012 to 2018 from the annual mean composite of VIIRS dataset by applying Otsu's method. The threshold of the lit area was found to be between 19.25 to 20.93 nW cm$^{-1}$sr$^{-1}$. The samples of built-up area, sum of light intensity, and area of lit maps extracted from remotely sensed data are presented in Figure 3.

### 2.3. Data Preparation for Machine Learning Models

The collected social and economic databases contain unnecessary information, a wide range of missing values and noises, and inconsistencies for prediction models. Before conducting any statistical analysis, the cleansing process, such as eliminating irregularities and outlying data, is necessary to ensure meaningful data and the reliability of model outputs [35]. Our cleaning method employed RapidMiner® Software by using an imputing operator to estimate missing values through learning models. Then, we normalized all variables using the statistical method (z-score). The equation is shown in Equation (2).

$$x' = \frac{x - \mu}{\sigma},$$

(2)

where $x'$ is normalized $x$, $\mu$ is the mean, and $\sigma$ equals the standard deviation. The z-score is a common normalization technique. It preserves the original distribution of the data and is less influenced by outliers. After the cleansing and normalizing processes were completed, the descriptive statistics of each variable were calculated (Table 1).

**Table 1.** Descriptive statistics of variables.

|  | Mean | STD. Error | SD. | Min | Max |
|---|---|---|---|---|---|
| Total Electricity Consumption [1] (TEC) | 847.98 | 78.60 | 1054.48 | 6.73 | 4293.21 |
| Population (TP) | 75,633.17 | 5003.59 | 67,130.15 | 4580.00 | 294,682.00 |
| Household (TH) | 38,892.08 | 3224.76 | 43,264.70 | 1903.00 | 199,851.00 |
| Sum of Light [2] (SL) | 6682.55 | 674.82 | 9053.65 | 49.40 | 41,146.70 |
| Area of Lit [3] (AL) | 50.75 | 5.32 | 71.34 | - | 324.81 |
| Built up Area [3] (BU) | 79.51 | 4.92 | 66.05 | 1.74 | 264.27 |
| Population Density (PD) | 216.80 | 15.97 | 214.25 | 32.00 | 1130.00 |
| Area [3] | 445.46 | 21.79 | 292.34 | 7.07 | 1422.90 |
| Year | 2014.50 | 0.13 | 1.71 | 2012 | 2017 |
| Gross Domestic Product [4] (GDP) | 592,073.89 | 57,014.05 | 764,923.78 | 3265.83 | $4.08 \times 10^6$ |

Unit: [1] GWh; [2] nW cm$^{-1}$sr$^{-1}$; [3] km$^2$; [4] million Baht.

Next, a data splitting strategy was applied to randomly split the available dataset into three groups for the ML tasks (training, validating, and testing) [36,37]. We split the data from the 2012–2018 dataset (100%: $n = 210$) into three sets: (1) training (60%: $n = 126$), (2) validating (20%: $n = 42$), and (3) testing (20%: $n = 42$). The training set was used for learning and estimating parameters of the model, while the validation set was used to evaluate the model and for model selection. Lastly, the testing set was a set of examples used to assess the model's predictive performance.

### 2.4. ML Model Setting

ML-based prediction models consist of four types of resources: dataset, training process, model, and prediction. The workflow was to use training data during the training process for model creation. This model with data input was used to establish the predictions. The activation functions were first employed for linear and nonlinear models to train an ML model, depending on the specific ML-based prediction models. Therefore, the ML was trained by using simple frameworks. The tweaking parameters as hyperparameter optimization were applied to improve the trained ML by adjusting model parameters to achieve the best results. In addition, for ML model construction, the feature selection method was necessary for identifying a subset of only relevant input variables to be used. The variable selection process was beneficial in reducing the overfitting and training time while improving accuracy [38]. In this study, we experimented based on two assumptions: (1) the electricity consumption prediction model is linear, the variables are of a normal distribution, and (2) there is non-linearity in the variables. Hence, we applied Multiple Regression (MR), Decision Tree (DT), and Support Vector Regression (SVR) to deal with linear and nonlinear functions. The above process was conducted in RapidMiner® Software in five steps (see Appendix A Figure A1).

Step one: The basic processing is composed of three tasks, including (1) primary pre-processing data, which started from loading the dataset and performed some basic pre-processing tasks such as defining target columns, removing unnecessary columns, and unifying value types, and then filtering examples. (2) The next task was creating a training and validation set by splitting data into a 60:40 ratio for performance calculation. (3) Then, the essential feature engineering was applied to replace missing and infinite value. This also includes data encoding and removing columns with too many nominal values.

Step two: The automatic feature selection and modeling. The feature selection was automatically performed to arrange and optimize the number of predictor variables used

by each model. This feature selection supports multi-objective feature selection and defends a balance correlation value between 1 and 0. The multi-objective evolutionary algorithm was operated to find the best feature sets to build models for the feature selection process. Each feature set was Pareto-optimal concerning complexity vs. model validation. The complexity was calculated based on the feature set where each feature in the set contributes complexity one. The selected feature sets were determined based on the optimal trade-offs between complexity and model errors. Afterward, we trained the models and applied automatic hyperparameter tuning (parameter optimization).

Step three: The transform validation and scoring data were performed, preparing data for validation and the scoring process in the next step.

Step four: Scoring and validation processed by applying the trained models on validation and scoring dataset. In this step, we applied automatic hyperparameter tuning, such as optimizing parameters with three k-folds validation, to estimate the statistical performance of a learning model. Then, we calculated MAE and RMSE to determine the performance of prediction models.

Step five: The final prediction models were created by training models with the same parameters on the combined training and validation dataset.

### 2.4.1. Multiple Regression (MR)

Multiple Regression is a statistical technique of linear regression that predicts the value of a response (dependent) variable based on two or more explanatory (independent) variables. It is one of the most commonly used statistical techniques and has been adopted in many research studies that require continuous output results [18]. The multiple regression equation was shown in the following form:

$$y = b_1 x_1 + b_2 x_2 + \ldots + b_n x_n + c \tag{3}$$

where $y$ denotes the total electricity consumption (TEC), and each $x_i$ is the feature selected from the statistical variables in Table 1, such as total population (TP), household (TH), sum of light (SL), etc. $b_i$ (i = 1, 2 ... $n$) are the regression coefficients. $c$ is the independent identically distributed normal error, representing the deviation of the predicted value from the observed value of TEC.

The fundamental assumptions of MR include the linear relationship between TEC and the selected features (e.g., total population (TP), total household (TH), sum of light (SL)). The MR model parameters are often estimated with the training data using the ordinary least squares approach by minimizing the sum of differences between predicted and observed values. After the model is built, it can be validated and assessed with the testing dataset to measure the accuracy of the prediction models. The coefficient of determination ($R^2$) is often used to measure how much of the variation in the dependent variable can be explained by independent variables. In this study, we inputted the normalized variable to RapidMiner® Software. Then, the automatic feature selection function was applied to select the optimized prediction variables for the best fitting model.

### 2.4.2. Decision Tree (DT)

A Decision Tree is a non-parametric supervised learning method that repeatedly divides a set of training data into smaller subsets based on tests of one or more variables to create a prediction model for the target value [39]. Unlike many other statistical approaches, the DT (regression tree) does not have the requirements of value distribution or the independence of the variables from one another [40]. The model is trained by learning based on decision rules from features. The collection of nodes intends to decide on values connected to estimate a numerical target value (regression approach). The selected variable will optimally separate to reduce the error following the criterion. The new nodes are repeatedly built until the estimated numerical value meets the target (criteria).

The sampling of new nodes is repeated until the stopping criteria are met. A prediction for the class label "attribute" is determined depending on the majority of "examples". The

criteria reach this leaf during generation, while an estimation for a numerical value is obtained by averaging the values in a leaf.

Our DT model was built by determining optimal parameters using threefold cross-validation. We then trained by setting the splitting criterion as Least Square (LS) error, which minimizes the squared distance between the average values in the node regarding the actual value. The LS is one of the standard methods to build a regression tree. It can improve the computational efficiency result that deals with the nominal and numerical variable tree [41]. The least square error criterion can be expressed as follows:

$$\frac{1}{n}\sum_{i}^{n}(y_j - r(\beta, x_i))^2 \tag{4}$$

where $n$ is the sample size, $x_i$ is a data point, and $r(\beta, x_i)$ is a regression model for each pair of $(x_i, y_i)$. The model criterion was set for 20 maximum depths. We also applied auto-prepruning to control the parameters of minimal gain = 0.05, minimal leaf size = 2, minimal size for split = 4, and a number of prepruning alternatives = 3 were used as stopping criteria.

### 2.4.3. Support Vector Regression (SVR)

A Support Vector Regression (SVR) is a non-parameter supervised machine learning model. The SVR function is derived based on training data to predict numerical values that find the best fitting line in the hyperplane with the maximum number of points [42]. The training attributes are defined by feature vectors $x$ selected from the set of all features $X$ (as in Table 1) and numerical label $y \in R$, which represents the prediction of TEC [43]. SVR is a regression function that is linear to the kernel function $K(x_i, x)$, trying to capture the non-linear relationship between selected features. Specifically, the prediction function is expressed as:

$$\sum_{i \in SV}(\alpha_i - \alpha_i^*)K(x_i, x) + b \tag{5}$$

where $x$ denotes TEC, $x_i$ represents the prediction variables in Table 1 (such as TP, AL, SL, Bu, and GDP), $\alpha_i$ and $\alpha_i^*$ are the Lagrange multipliers, and $b$ is an intercept. This kernel function $K(x_i, x)$ was set up to map the nonlinear function to the flattest function in a feature space. We also applied the polynomial kernel, which is suitable for the normalized training dataset. The polynomial kernel is defined by:

$$K(x_i, x) = (1 + (x_i \cdot x))^d \tag{6}$$

where $d$ is the polynomial degree, and the kernel degree parameter specifies it. We input the normalized variables into RapidMiner® Software for this study following five steps of building the prediction model. We applied the radial kernel with gamma = 0.001 and cache = 200. The hyperparameter C, which controls the relaxation of the error minimization problem, was set as 100. The convergence epsilon was set at 0.001 with a max iteration at 100,000. SVR constructed a hyperplane or set of hyperplanes in a high- or infinite-dimensional space, which can be used for regression prediction with a fast algorithm and sufficient results.

### 2.5. Model Evaluation and Validation

To determine the accuracy and appropriate selection of the prediction method, we used two statistical measures, (1) mean absolute error (MAE) and (2) root mean square error (RMSE). A mean absolute error (MAE) measures the average magnitude of the errors in a set of predictions without considering their direction and the average of absolute differences between prediction and actual observation over the test samples with equal weight on all individual differences (Equation (7)) [19].

$$MAE = \frac{1}{n}\sum_{j=1}^{n}|y_j - \hat{y}_j| \tag{7}$$

where $n$ denotes the number of samples. $|\hat{y}_i - y_i|$ are the absolute residuals between the actual values and the predicted values, where $y_i$ is the observed value for the observation and $\hat{y}_i$ is the predicted value.

This research also applied a root mean square error (RMSE) to measure the standard deviation of residuals. RMSE is the standard deviation of the prediction errors (residuals) and can measure residual dispersal. A basic formula of RMSE is shown below:

$$RMSE = \sqrt{\frac{\sum_{i=1}^{n}(\hat{y}_i - y_i)^2}{n}} \tag{8}$$

## 3. Results

### 3.1. Variable Correlation

The Pearson correlation coefficients were calculated to determine the correlation among variables. The result is illustrated in Table 2. Total electricity consumption (TEC), a dependent variable in this study, appears highly correlated with sum of light intensity (SL), area of lit (AL), and built-up (BU) area. Moreover, the top three correlated variables are also highly correlated among themselves. The total population and household showed moderate correlation coefficients to TEC, while other variables had low correlation values (see Figure 4). The last row in Table 2 presents the weight of each variable to the prediction (dependence variable) by using a correlation calculation. The higher the weight of an attribute, the more relevant it is considered. These results helped the variables (feature) selection process to find the optimal variables, building the models by automatically selecting the appropriate candidate features by their information gain concerning the prediction results [44].

**Table 2.** The correlation coefficients between variables.

|      | Year | TP   | TH   | SL   | AL   | BU   | PD    | Area | GDP  | TEC |
|------|------|------|------|------|------|------|-------|------|------|-----|
| Year | 1    |      |      |      |      |      |       |      |      |     |
| TP   | 0.10 | 1    |      |      |      |      |       |      |      |     |
| TH   | 0.14 | 0.95 | 1    |      |      |      |       |      |      |     |
| SL   | 0.12 | 0.75 | 0.83 | 1    |      |      |       |      |      |     |
| AL   | 0.04 | 0.79 | 0.87 | 0.96 | 1    |      |       |      |      |     |
| BU   | 0.03 | 0.84 | 0.84 | 0.89 | 0.89 | 1    |       |      |      |     |
| PD   | 0.00 | 0.65 | 0.64 | 0.34 | 0.46 | 0.38 | 1     |      |      |     |
| Area | 0.00 | 0.17 | 0.09 | 0.13 | 0.04 | 0.22 | −0.32 | 1    |      |     |
| GDP  | 0.07 | 0.68 | 0.75 | 0.97 | 0.90 | 0.83 | 0.27  | 0.14 | 1    |     |
| TEC  | 0.08 | 0.56 | 0.63 | 0.83 | 0.78 | 0.77 | 0.15  | 0.12 | 0.83 | 1   |

### 3.2. Variables Selection

An automatic feature selection process was used to tune down the number of predictor variables employed by the built models. Adjusting the number of parameters is a significant part of the model building process, which helps improve the processing time of each prediction model. RapidMiner® Auto Model's Feature selection used a multi-objective evolutionary algorithm to find the best predictor variables (feature sets) concerning the trade-off between complexity and model accuracy [45]. The complexity was determined based on the set of features where each variable in the set contributes complexity. The result of each feature set is Pareto-optimal concerning these two dimensions (complexity and model accuracy). In this study, each model selected a specific feature set using an automatic feature selection process. The result of the feature selection process (predictors for each model) is illustrated in Table 3.

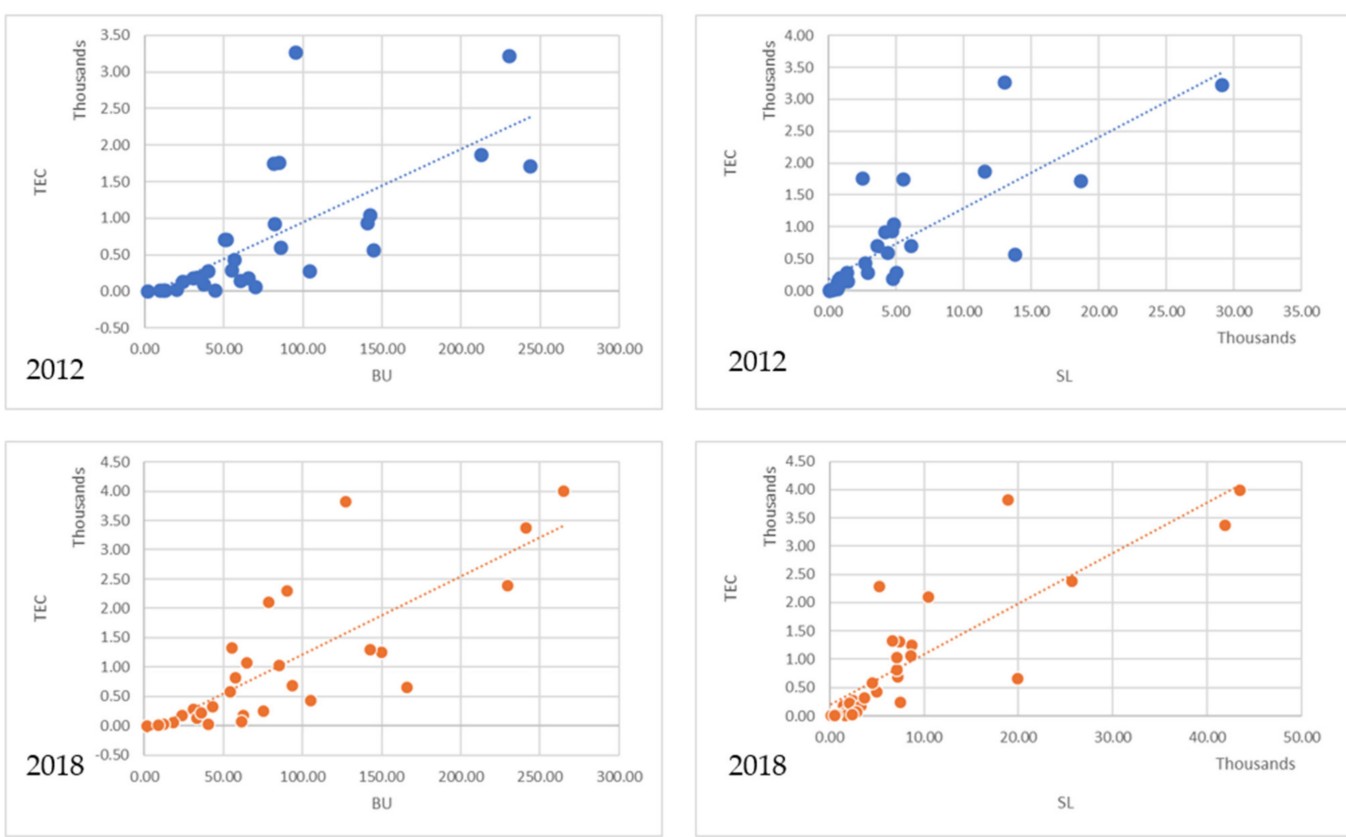

**Figure 4.** The scatter plot and correlation between predictors and prediction variable (sample of the year 2012 and 2018); (**a**) TEC and BU and (**b**) TEC and SL.

**Table 3.** Electricity consumption estimation predictors and their weight in each model.

| Variables / Models | Year | TP | TH | SL | AL | BU | GDP | PD | Area |
|---|---|---|---|---|---|---|---|---|---|
| Multiple Regression (MR) | 0.12 | | | | 0.86 | 0.49 | | | |
| Decision Tree (DT) | | | 0.51 | | | | 0.20 | | 0.004 | 0.09 |
| Support Vector Regression (SVR) | | 0.03 | 0.04 | 0.08 | | 0.06 | 0.05 | 0.06 | |

*3.3. Estimation Models and Validation*

The MR model is presented in Equation (9). The variables, including AL, BU, and area, were selected from the feature selection method. The MR model provided the lowest accuracy compared to DT and SVR (see Table 4).

$$TEC = 0.44\,(AL) + 0.37(BU) + 0.04(Year) + 0 \tag{9}$$

**Table 4.** The performance of prediction models.

| | Multiple Regression (MR) | Decision Tree (DT) | Support Vector Regression (SVM) |
|---|---|---|---|
| MAE | 0.33 | 0.12 | 0.20 |
| RMSE | 0.51 | 0.24 | 0.39 |
| R2 | 0.81 | 0.95 | 0.94 |

The DT model was created from five attributes (one target attribute, TEC, and four predictor attributes, BU, TH, PD, and area) and their decision label value (see Appendix B Table A1). Total household (0.393) was the highest weight attribute, followed by an area of lit (0.192) and population (0.005). Figure 5b illustrates the DT prediction chart, which led to the best performance in electricity prediction among three models (see Table 4) with an RMSE of 0.24, MAE of 0.12, MSE of 0.06, and the highest $R^2$ of 0.95. The 210 training datapoints trained the final SVR prediction model with four attributes: TP, TH, SL, BU, and PD (see Figure 5c). The kernel function of each variable was present by $w(x_i)$ in Figure 5c, and the offset is 0.655. As shown in Table 4, the SVR prediction model had the second-best prediction performance.

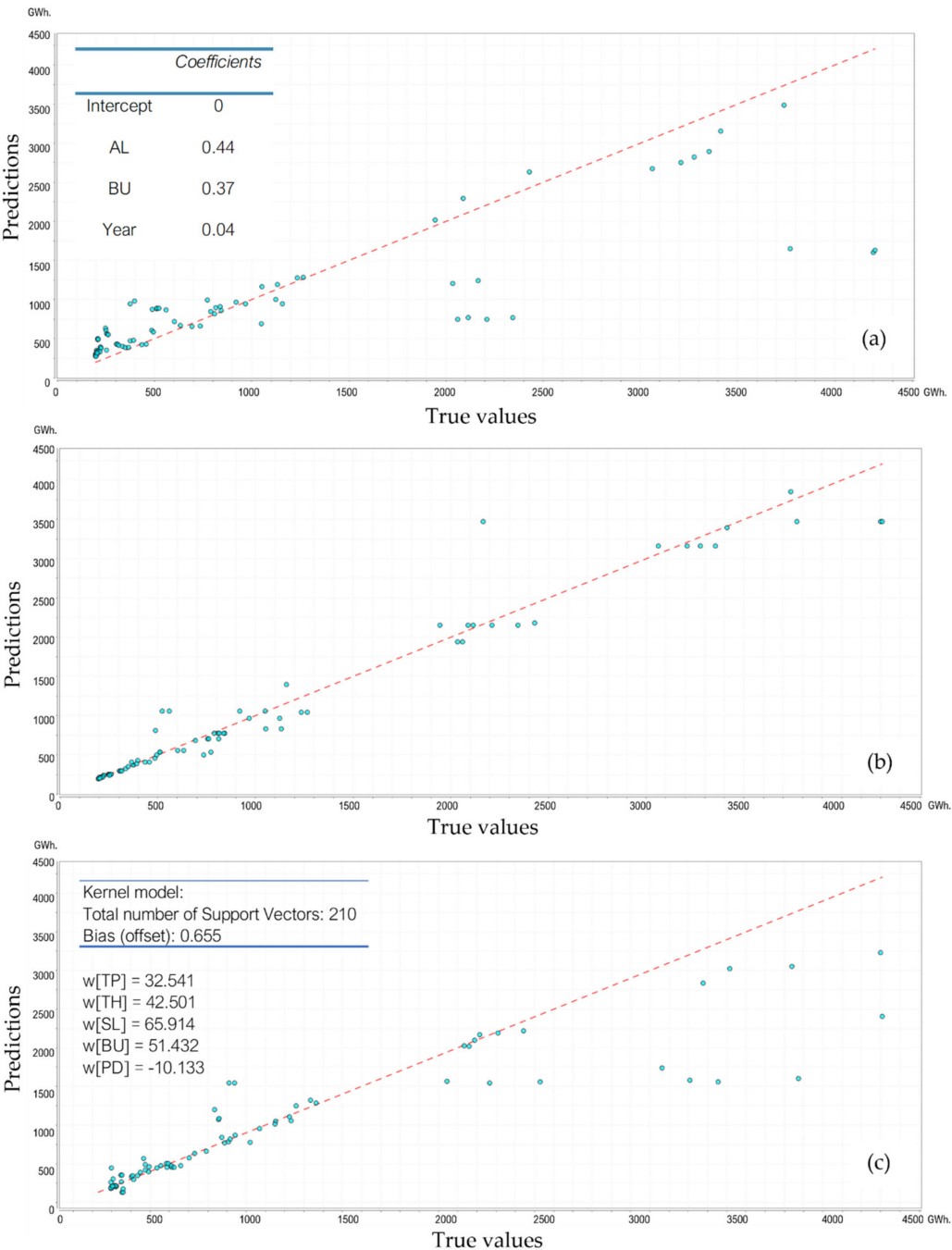

**Figure 5.** The scatter plots of predicted values versus actual values from different models, including (**a**) Multiple regression, (**b**) Decision Tree, and (**c**) Support Vector Regression.

A comparison of the prediction models' performance is illustrated in Table 4, in which DT provided the best performance with the lowest MAE, RMSE, and highest $R^2$, followed by SVR and MR. Therefore, DT achieved the highest accuracy and was chosen to estimate the district-level electricity consumption in Thailand's EEC.

## 4. Discussion

### 4.1. High Correlation of Geographic Variables to Electricity Consumption

This study applied the socioeconomic survey data (TH and TP) and extracted geographic data from remote sensing images (SL, AL, and BU). The extracted data from satellite images including SL, AL, and BU presented a very high correlation to electricity consumption ($0.77 \leq R \leq 0.83$) compared to socioeconomic data (TH and TP), which provided R = 0.6. However, the GDP also provided a high correlation to total electricity consumption at 0.83. These results suggest that the variables extracted from remote sensing products could estimate the electricity consumption at the district level. Table 2 illustrates that SL was highly correlated with TEC. Moreover, the built-up area was a common predictor, chosen by all models through the feature selection method (see Table 3). Hence, we conclude that the remotely sensed extraction data could estimate electricity consumption at a district level.

### 4.2. Remotely Sensed Data—High Performance and Ease of Extraction

The past study confirmed that the nighttime light retrieved from the satellite could quantify electricity consumption at a provincial scale [1]. Moreover, we found that the lit area, sum of light intensity, and built-up area could also estimate the electricity consumption at the district level. These geographic variables played a considerable role in prediction models by providing high weight (see Table 3). For example, the built-up area was the second-highest weight (0.2) in the DT prediction model, and the area of lit was the highest weight (0.86) in the MR prediction model. However, the built-up area was more complex in terms of extraction methodology, and it was challenging to control the accuracy of data extraction results. Figure 6 illustrated that the built-up area is highly correlated to the area of lit and sum of light intensity. For this reason, the nighttime light intensity was a better proxy indicator to estimate the electricity consumption at the district level.

### 4.3. Model Complexity and Accuracy, Key Caveats, and Limitations

There were various methods to estimate electricity consumption in past research. However, because of the limitation of the number of datasets, we selected three MLAs, including Multiple Regression (MR), Decision Tree (DT), and Support Vector Regression (SVR), to predict the district's electricity consumption. From Figure 5, it is clear that DT's predicted values showed the closest one-to-one relationship to the actual values, which indicates that the DT model provided the best performance with the lowest MAE, RMSE, and the highest R2. On the other hand, MR and SVR were more likely to underestimate the actual values with weaker goodness of fit than the DT model.

The regression method in machine learning may be challenged by the complexity of predictor variables, which could generate a high bias model. The increasing complexity is possibly caused by the decreasing of model accuracy. Thus, we need to optimize both complexity and accuracy, avoiding a more complex and over-fitting model. Figure 7 compares the trade-off between the model complexity and the error rate of MR, DT, and SVR models.

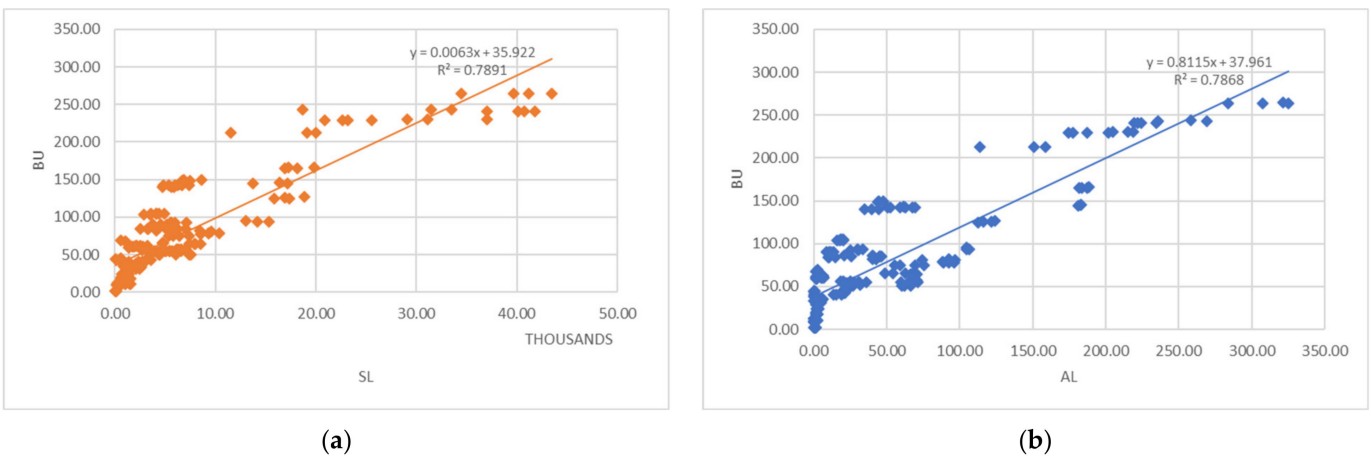

**Figure 6.** The correlation between geographical variables (remotely sensed data extraction); (**a**) BU and SL; (**b**) BU and AL.

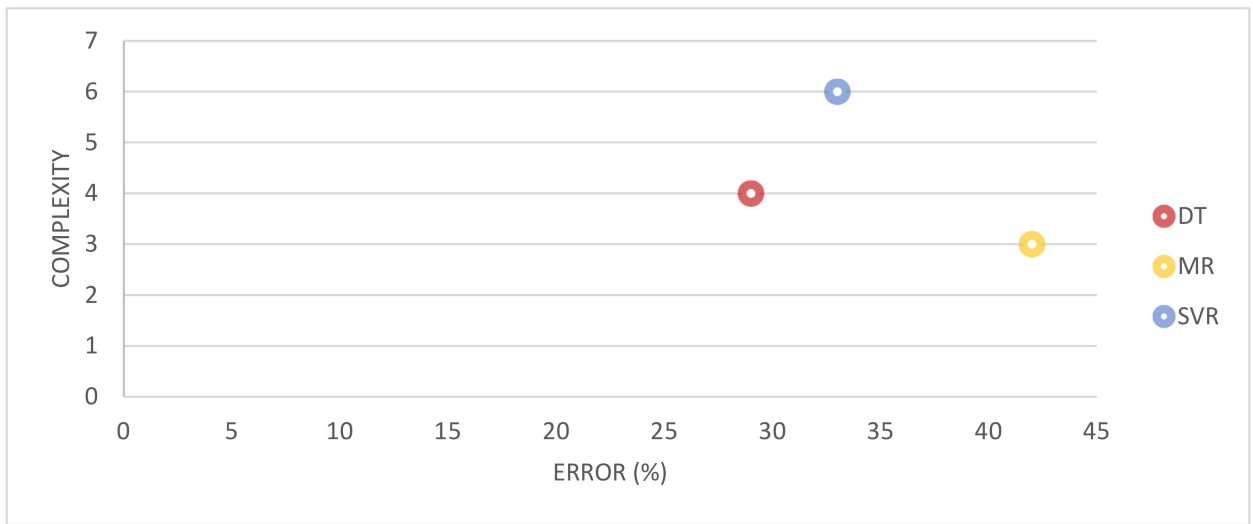

**Figure 7.** The optimal trade-offs between complexity and error of each estimation model.

The MR provided less complexity but the highest error, while DT provided the best performance with moderate complexity but the lowest error. In addition, the model parameter adjusting was a key to the predicting result. We let each model find the optimal parameters that provided the best training result. However, the prediction models may still need more adjusting in the details of each hyperparameter. The most critical hyperparameters are (1) stopping criteria and (2) pruning method, which control the balancing between model accuracy and complexity. Consequently, we need to balance complexity and accuracy, improving the model performance.

In addition, DT is a complex model which tends to overfit. We can avoid this problem by setting the constraints on model parameters and pruning. The constraints on tree size should be set by fine-tuning the hyperparameters such as (1) minimum samples for a node split and a terminal node, (2) maximum depth of the tree (vertical depth), and (3) maximum feature to consider for a split. The tree pruning should be optimized between the size of the learning tree (horizontal grow) and predictive accuracy, then measured by cross-validation set for best optimizing performance. However, based on the number of data points in this study, our DT model was a small tree with a lower risk of overfitting than a large tree. The overfitting problem should be considered when applying the model to a larger dataset in further study.

The model accuracy can be impacted by (1) the quality of the input dataset and (2) model adjusting parameters. Principally, the input data validation may be applied to improve the model accuracy. This study only validated the accuracy of remote sensing data extracting results of 2012 and 2016. We used 500 random sampling points from the national land-use dataset, verifying them with virtual ground truth from Landsat 7 and 8 images. In this work, the accuracy of extracting data for 2012 and 2016 was 78% and 85%, respectively, which was adequate for managing electricity on the district grid in EEC, Thailand, without the dwelling dataset. The more detailed validation of remote sensing data extraction should be conducted because the accuracy of the data extraction process will affect the result of the prediction model. However, if we can improve the extraction accuracy, the models should lead to more accurate predictions. In addition, we assumed that there are no invalid data in the socio-economic dataset following the standard of data source (The National Statistic Office, Thailand). However, this may not be true.

Despite the usefulness of this study's generated models and findings, we acknowledge that some caveats should be mentioned. First, this study's available data were somewhat limited due to data availability in both spatial and temporal domains. The data were available only from 2012 to 2018, covering a small number of districts (30 districts) in the study area. Further study still needs to utilize the hyperparameter tuning of the Decision Tree model in the following metrics: the total number of nodes, total number of leaves, tree depth, and number of attributes used to optimize model accuracy. Moreover, experimenting on different geographic characteristic areas will enhance understanding of the prediction model at the diverse remotely sensed dataset.

## 5. Conclusions

This study compared three estimation models for electricity consumption at the district level in Eastern Thailand and evaluated the usefulness of remotely sensed data extraction as well as a general socioeconomic dataset as model features. The result revealed that the Decision Tree (regression method) algorithm is the best-fitting model with the highest accuracy at the lowest MAE, RMSE, and fastest total time than the other models. The findings from this study also highlighted the benefits of utilizing remotely sensed extracted variables, including lit area, sum of light intensity, and built-up area. Specifically, the nighttime light data were significant for estimating electricity consumption at the district level. They can help understand human activity and reflect the economic status and its impact in the industrial parks and neighborhood areas. These data also represent an excellent proxy indicator to measure economic activity, which is valuable for developing countries with a limited budget and data, such as Thailand.

Nevertheless, the nighttime light data capture an artificial light that varies by the intensity of the source of light generated by the bulbs. LEDs have increasingly replaced regular light bulbs in the past decade because they require less energy and provide the same brightness. The changing of light bulb types may reflect the utilization of nighttime light intensity for estimating electricity consumption. Starting from 2015, Thailand's government has promoted LED usage within the 20 years of saving energy plan, which is still ongoing [46]. Thus, this paper, covering the period from 2012 to 2018, assumed little impact from the policy and has yet to consider the effect of different light sources.

**Author Contributions:** Conceptualization, S.H. and D.C.; Data curation, S.H.; Formal analysis, S.H.; Methodology, S.H. and D.C.; Supervision, D.C.; Writing—original draft, S.H.; Writing—review and editing, D.C. All authors have read and agreed to the published version of the manuscript.

**Funding:** This research was partially funded by the Natural Sciences and Engineering Research Council of Canada (NSERC) Discovery Grant.

**Institutional Review Board Statement:** Not applicable.

**Informed Consent Statement:** Not applicable.

**Acknowledgments:** The authors would like to express their gratitude to the Provincial Electricity Authority (PEA), Geoinformatics and Space Development Agency (GISTDA), National Statistical Office (NSO), Office of the National Economic and Social Development Council (NESDC), and Ministry of Higher Education, Science, Research, and Innovation (MHESI), Thailand, for data and research funding. We also highly appreciate Nuntikorn Kitratporn and Narut Soontranon, who dedicated valuable time to proofreading the manuscript.

**Conflicts of Interest:** The authors declare no conflict of interest.

# Appendix A

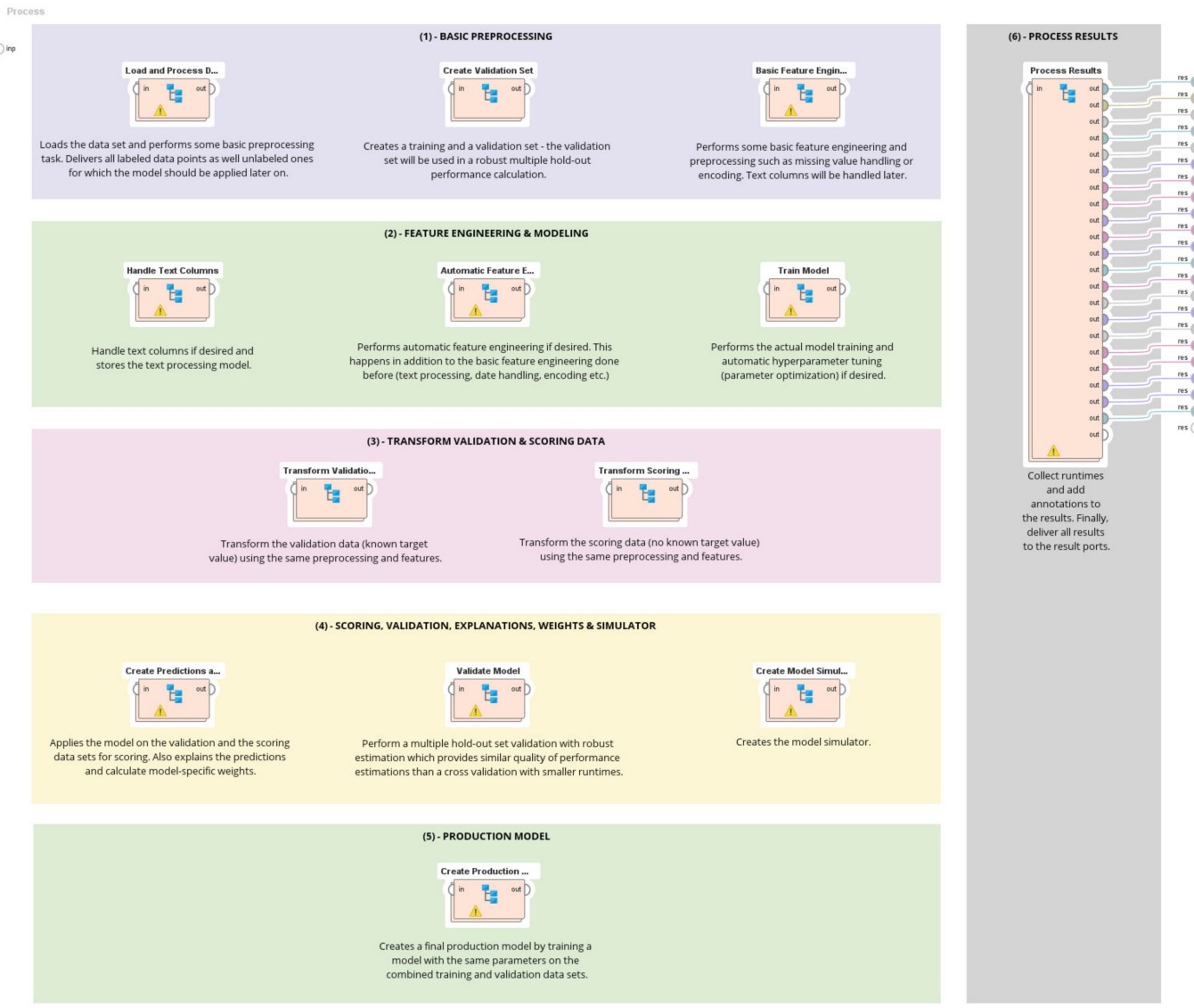

**Figure A1.** The process of creating a Decision Tree from RapidMiner® Software.

## Appendix B

**Table A1.** The description of the Regression Tree Model (Decision Tree).

```
TH > −0.148
| BU > 2.255
| | BU > 2.619: 2.732 {count = 4}
| | BU ≤ 2.619
| | | PD > 0.317
| | | | BU > 2.419
| | | | | PD > 0.417: 2.385 {count = 2}
| | | | | PD ≤ 0.417: 2.301 {count = 3}
| | | | BU ≤ 2.419: 2.072 {count = 3}
| | | PD ≤ 0.317: 1.505 {count = 2}
| BU ≤ 2.255
| | PD > −0.441
| | | BU > 1.644
| | | | TH > 1.683: 1.340 {count = 4}
| | | | TH ≤ 1.683: 1.029 {count = 3}
| | | BU ≤ 1.644
| | | | TH > 0.011
| | | | | Area > −0.591
| | | | | | PD > −0.374
| | | | | | | PD > 0.524
| | | | | | | | BU > 0.510: 0.320 {count = 6}
| | | | | | | | BU ≤ 0.510: 0.076 {count = 7}
| | | | | | | PD ≤ 0.524: 0.857 {count = 2}
| | | | | | PD ≤ −0.374
| | | | | | | TH > 0.034: −0.160 {count = 2}
| | | | | | | TH ≤ 0.034: −0.190 {count = 2}
| | | | | Area ≤ −0.591: −0.217 {count = 9}
| | | | TH ≤ 0.011
| | | | | PD > −0.008
| | | | | | BU > −0.133
| | | | | | | TH > −0.084
| | | | | | | | BU > 0.113: 1.192 {count = 3}
| | | | | | | | BU ≤ 0.113: 1.038 {count = 5}
| | | | | | | TH ≤ −0.084
| | | | | | | | PD > 0.282: 0.904 {count = 2}
| | | | | | | | PD ≤ 0.282: 0.836 {count = 2}
| | | | | | BU ≤ −0.133: 0.108 {count = 3}
| | | | | PD ≤ −0.008
| | | | | | BU > 0.644
| | | | | | | TH > −0.058: 0.256 {count = 2}
| | | | | | | TH ≤ −0.058: 0.150 {count = 2}
| | | | | | BU ≤ 0.644: −0.218 {count = 5}
| | PD ≤ −0.441
| | | TH > 0.176
| | | | PD > −0.534: 3.203 {count = 2}
| | | | PD ≤ −0.534
| | | | | TH > 0.436: 2.740 {count = 2}
| | | | | TH ≤ 0.436: 2.473 {count = 2}
| | | TH ≤ 0.176: 1.705 {count = 2}
TH ≤ −0.148
| TH > −0.306
| | PD > −0.369
| | | PD > −0.302
```

**Table A1.** *Cont.*

```
| | | | PD > 0.117: −0.089 {count = 3}
| | | | PD ≤ 0.117: 0.046 {count = 2}
| | | PD ≤ −0.302: −0.459 {count = 2}
| | PD ≤ −0.369
| | | PD > −0.388: 0.402 {count = 2}
| | | PD ≤ −0.388
| | | | PD > −0.397: 0.168 {count = 2}
| | | | PD ≤ −0.397: 0.057 {count = 3}
| TH ≤ −0.306
| | BU > −0.877
| | | Area > 0.778
| | | | BU > −0.632
| | | | | Area > 2.736
| | | | | | TH > −0.534: −0.784 {count = 3}
| | | | | | TH ≤ −0.534: −0.791 {count = 4}
| | | | | Area ≤ 2.736
| | | | | | TH > −0.425: −0.741 {count = 3}
| | | | | | TH ≤ −0.425
| | | | | | | PD > −0.690: −0.749 {count = 2}
| | | | | | | PD ≤ −0.690: −0.752 {count = 2}
| | | | BU ≤ −0.632
| | | | | TH > −0.566: −0.691 {count = 4}
| | | | | TH ≤ −0.566: −0.708 {count = 3}
| | | Area ≤ 0.778
| | | | TH > −0.391
| | | | | BU > −0.302
| | | | | | TH > −0.323: −0.488 {count = 2}
| | | | | | TH ≤ −0.323: −0.564 {count = 3}
| | | | | BU ≤ −0.302
| | | | | | TH > −0.363
| | | | | | | TH > −0.331: −0.261 {count = 2}
| | | | | | | TH ≤ −0.331: −0.189 {count = 2}
| | | | | | TH ≤ −0.363: −0.281 {count = 2}
| | | | TH ≤ −0.391
| | | | | BU > −0.629
| | | | | | BU > −0.325
| | | | | | | PD > −0.190
| | | | | | | | TH > −0.533: −0.587 {count = 2}
| | | | | | | | TH ≤ −0.533: −0.610 {count = 2}
| | | | | | | PD ≤ −0.190
| | | | | | | | BU > −0.280: −0.637 {count = 5}
| | | | | | | | BU ≤ −0.280: −0.658 {count = 3}
| | | | | | BU ≤ −0.325
| | | | | | | Area > −0.759
| | | | | | | | BU > −0.382: −0.387 {count = 5}
| | | | | | | | BU ≤ −0.382: −0.311 {count = 3}
| | | | | | | Area ≤ −0.759
| | | | | | | | BU > −0.575: −0.506 {count = 4}
| | | | | | | | BU ≤ −0.575: −0.541 {count = 3}
| | | | | BU ≤ −0.629
| | | | | | TH > −0.696
| | | | | | | PD > −0.641
| | | | | | | | TH > −0.483: −0.604 {count = 2}
| | | | | | | | TH ≤ −0.483: −0.625 {count = 5}
| | | | | | | PD ≤ −0.641
| | | | | | | | PD > −0.751: −0.571 {count = 4}
| | | | | | | | PD ≤ −0.751: −0.611 {count = 3}
```

**Table A1.** *Cont.*

```
| | | | | | TH ≤ −0.696
| | | | | | | TH > −0.703: −0.650 {count = 3}
| | | | | | | TH ≤ −0.703: −0.673 {count = 4}
| | BU ≤ −0.877
| | | BU > −0.972
| | | | TH > −0.650: −0.755 {count = 3}
| | | | TH ≤ −0.650: −0.776 {count = 4}
| | | BU ≤ −0.972
| | | | BU > −1.019
| | | | | TH > −0.772
| | | | | | TH > −0.768: −0.779 {count = 3}
| | | | | | TH ≤ −0.768: −0.785 {count = 2}
| | | | | TH ≤ −0.772: −0.792 {count = 2}
| | | | BU ≤ −1.019
| | | | | TH > −0.796
| | | | | | BU > −1.065
| | | | | | | TH > −0.640: −0.796 {count = 3}
| | | | | | | TH ≤ −0.640
| | | | | | | | Area > −0.527: −0.797 {count = 4}
| | | | | | | | Area ≤ −0.527: −0.797 {count = 3}
| | | | | | BU ≤ −1.065
| | | | | | | BU > −1.069: −0.794 {count = 2}
| | | | | | | BU ≤ −1.069: −0.796 {count = 2}
| | | | | TH ≤ −0.796
| | | | | | TH > −0.815: −0.802 {count = 3}
| | | | | | TH ≤ −0.815
| | | | | | | PD > 2.167: −0.803 {count = 2}
| | | | | | | PD ≤ 2.167: −0.803 {count = 2}
```

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
