# Peer review of "Estimating District-Level Electricity Consumption Using Remotely Sensed Data in Eastern Economic Corridor, Thailand"

_remotesensing, doi:10.3390/rs13224654_

Round 1
Reviewer 1 Report
This study proposed an electricity consumption estimation model at the district level using machine learning with publicly available statistical data and Built-up area (BU), area of lit (AL), and the sum of light intensity (SL) data extracted from Landsat and Suomi NPP satellite nighttime light images. The models created from three machine learning algorithms which included Multiple Linear Regression (MR), Decision Tree (DT), and Support Vector Regression (SVR), were compared. I consider that the sample size of this article may not enough, and the method and description of the experiment need to be improved.The comments are listed:
Comment 1:
In the Introduction, the scientific contribution of this research should be directly stated, in addition to listing the methods used in this article, the innovation and significance of this research should be directly explained. Frankly speaking, I have not seen enough scientific contributions from the introduction and subsequent chapters, and the combination with remote sensing is also far-fetched.
Comment 2:
Line 101-line 133
This paragraph describes the significance of the research. I do not think it should appear in the Materials and Methods section. The author may consider adding a Related Works section or putting this part in the introduction. The author mainly introduces the situation of Thailand’s economy and electricity consumption, etc., but the innovation of methods and sufficient workload should be reflected in the scientific research papers.
Comment 3:
Line 152- line 160
The mentioned remote sensing data should be introduced in more detail, and the reasons for choosing them should be explained. The author should state more details about the reasons for the choice, such as why Otsu's method was chosen.
Comment 4:
Line 377, the Figure 5. (c), the resolution and font of the text in the figure are not uniform with (a)
Comment 5:
The decision tree model is prone to produce an overly complex model, and the generalization performance of such a model to the data will be poor. This is the so-called over-fitting. Some strategies such as pruning, setting the minimum number of samples required for leaf nodes or setting the maximum depth of the number are the most effective ways to avoid this problem. But this is not reflected in this article, it is recommended to mention it in the discussion section
Comment 6:
As a predictive model with the best effect in this research, the author should explain how to avoid the shortcomings of decision trees and what kind of work has been done to prevent overfitting.
Comment 7:
Line 197, validation set and test set should describe their respective numbers and proportions, not put them together.
Author Response
Dear Reviewer 1,
I had revised the manuscript, Please see the attachment.
Best Regards,

Reviewer 2 Report
Dear Authors,
The paper “Estimating District-level Electricity Consumption Using Remotely Sensed Data in Eastern Economic Corridor, Thailand” is certainly of great quality and interest to the readers.
The reviewer has the following minor comments:
- It is not clear if cloud mask was applied or not to DNB VIIRS data. Please clarify.
- Page 5 which spectral bands of Landsat 7 and 8 were used?
- How the threshold value was calculated by using Otsu's method (the lit area was brightened than 20.93 174 ?? ??−1 ??−1 .) It depends on image and may be different for each image.
- How many images (Landsat and VIIRS) were used? Statistics over months or seasons would be interesting to plot.
- Figure 5 (a) and (c) and Figure 6 show some non-linearity (similar to logarithm). Please explain it.
- Figure 5 (b) Vertical axis title has to be fixed.
- The night light is only one indicator of electricity consumption. Please note that during last decade there is a tendency to use LED light sources instead of incandescent bulbs. The reviewer suggests to mention it.
Sincerely,
Reviewer
Author Response
Dear Reviewer 2,
I had revised the manuscript, Please see the attachment.
Best Regards,

Round 2
Reviewer 1 Report
Thanks for taking my advices